# Epidemiological Impact of Myocarditis

**DOI:** 10.3390/jcm10040603

**Published:** 2021-02-05

**Authors:** Ainoosh Golpour, Dimitri Patriki, Paul J. Hanson, Bruce McManus, Bettina Heidecker

**Affiliations:** 1Campus Benjamin Franklin, Charite Universitätsmedizin Berlin, 12203 Berlin, Germany; ainoosh.golpour@charite.de; 2Department of Medicine, Cantonal Hospital of Baden, 15005 Baden, Switzerland; Dimitri.Patriki@ksb.ch; 3Department of Pathology & Laboratory Medicine, University of British Columbia, Vancouver, BC V5K0A1, Canada; Paul.Hanson@hli.ubc.ca (P.J.H.); Bruce.McManus@hli.ubc.ca (B.M.)

**Keywords:** myocarditis, epidemiology, incidence and prevalence, myocarditis associated with COVID-19, etiology, diagnosis, regional differences

## Abstract

Myocarditis is an inflammatory disease of the heart muscle with a wide range of potential etiological factors and consequently varying clinical patterns across the world. In this review, we address the epidemiology of myocarditis. Myocarditis was considered a rare disease until intensified research efforts in recent decades revealed its true epidemiological importance. While it remains a challenge to determine the true prevalence of myocarditis, studies are underway to obtain better approximations of the proportions of this disease. Nowadays, the prevalence of myocarditis has been reported from 10.2 to 105.6 per 100,000 worldwide, and its annual occurrence is estimated at about 1.8 million cases. This wide range of reported cases reflects the uncertainty surrounding the true prevalence and a potential underdiagnosis of this disease. Since myocarditis continues to be a significant public health issue, particularly in young adults in whom myocarditis is among the most common causes of sudden cardiac death, improved diagnostic and therapeutic procedures are necessary. This manuscript aims to summarize the current knowledge on the epidemiology of myocarditis, new diagnostic approaches and the current epidemiological impact of the COVID-19 pandemic.

## 1. Introduction

Myocarditis is a heterogeneous disease with a variety of symptoms ranging from mild chest discomfort to cardiogenic shock. As it often mimics other common disorders, such as coronary artery disease (CAD), diagnosis may be challenging [1]. Over the years, various diagnostic tests have evolved to identify patients suffering from myocarditis. Nowadays, a diagnosis of myocarditis includes clinical, laboratory, imaging and histological parameters. While the gold standard for diagnosis of myocarditis is endomyocardial biopsy (EMB) [2,3], cardiac magnetic resonance imaging (CMR) is considered a non-invasive alternative in patients with suspected myocarditis [3,4,5]. New diagnostic algorithms and tools help us better understand its true proportions and indicate a potential underestimation of the prevalence of myocarditis. This is reflected in the discrepancy within the published data about the prevalence of myocarditis ranging from 10.2 to 105.6 per 100,000 people worldwide [6]. The potentially severe complications range from heart failure to sudden cardiac death [3].

Most cases of myocarditis are caused by infectious agents, toxic substances, drugs or autoimmune disorders [7]. Hence, it is increasingly recognized that myocarditis is an inflammatory condition of the myocardium triggered by various factors rather than a distinct cardiovascular disease.

This review aims to shed light on the worldwide epidemiological impact of myocarditis and provide perspective on the apparent contribution of heart damage associated with COVID-19 observed during the ongoing pandemic.

## 2. Etiology of Myocarditis

### 2.1. Infection

Acute lymphocytic myocarditis is the most common type of myocarditis and is frequently caused by pathogens, such as adenoviruses, after a respiratory infection [8,9]. A study including 12,747 routine autopsies over a period of 10 years in a general population revealed that lymphocytic myocarditis was present in 1.06% of the cases [10].

However, the reported incidence of lymphocytic myocarditis confirmed by EMB varies in the literature, in part due to differences in diagnostic sensitivity of methods used for histology or sample collection.

In 1989, Chow and colleagues determined that it required up to 17 right ventricular biopsy specimens to detect 79% of cases with myocarditis—a number that is unrealistic to achieve in the clinical setting [11]. Thus, it is expected that when standard biopsy techniques are used, a low sensitivity of detection will be achieved. This is, of course, a different issue than the usually patchy nature of myocarditis detected, which will be addressed below.

In 2010, Dec described the incidence of lymphocytic myocarditis to be 55% among cases of biopsy-proven myocarditis [12]. A subsequent study identified lymphocytic myocarditis in most of the patients (95.5%) diagnosed with myocarditis based on EMB. In that study, 564 out of 1752 patients diagnosed with myocarditis received left ventricular EMB, and 1118 of them received biventricular EMB. Biventricular EMB increased diagnostic sensitivity in this study [13]. The high degree of variability in diagnosis of myocarditis via EMB may be attributable to intra/interobserver error, but is more likely a reflection of the heterogeneous, focal patchy nature of the disease. Hence, in contrast to post-mortem gross observation, EMB would require an unrealistic sample size of biopsy pieces to detect regions of inflammation and injury that would accurately represent the true prevalence of myocarditis among patients with clinical symptoms of the disease.

It is generally difficult to determine the true incidence of myocarditis because, on the one hand, as described above, the disease has a patchy nature, and on the other hand, EMB, the gold standard for the diagnosis of myocarditis, is not frequently performed because of its invasive character [2,3]. In 1986, Dallas criteria were established with the aim of providing a standardized histopathological categorization that could be helpful for the diagnosis of myocarditis using EMB [14]. However, interpretation of the histological data, as well as sampling errors of EMB itself, further impedes final diagnosis [4,5]. Nowadays, electroanatomic mapping (EAM) is an emerging alternative approach for the diagnosis of myocarditis, which could lead to the improvement of sensitivity and the reduction in false-negative rates in endomyocardial biopsy (EMB) [15]. Another non-invasive diagnostic alternative is cardiac magnetic resonance (CMR) imaging. However, access to CMR may be limited, and the lack of financial coverage of CMR by many insurance companies still creates a challenge in many countries. Differences in the availability and expertise of diagnostic tools and procedures in the various regions may also have contributed to differences in the reported prevalence (Figure 1).

According to the World Health Organization (WHO), in the period from 1975 to 1985, *Coxsackievirus B3* (*CVB3*, an enterovirus) was the most common virus causing myocardial injury worldwide (34.6 per 1000 cases), followed by *influenza B virus* (17.4 per 1000 cases), *influenza A* (11.7 per 1000 cases), *Coxsackie virus A* (9.1 per 1000 cases) and *cytomegalovirus* (*CMV*) (8.0 per 1000 cases) [16]. Bowles and colleagues as well as Martin and co-workers demonstrated that *adenoviruses* were predominant causes of myocarditis in children [17,18].

In European studies, *Parvovirus B19* (*PVB19*) was the most common virus found in patients suffering from myocarditis [19]. However, there is ongoing debate whether *PVB19* persistence in the myocardium affects clinical outcomes or whether it is in certain cases an “incidental finding” or “bystander” present on myocardial biopsies [20,21,22]. Most EMB samples from patients with acute myocarditis or inflammatory cardiomyopathy show low copy numbers of *PVB19* DNA. However, only the presence of >500 viral DNA copies of *PVB19* per microgram of cardiac DNA is currently associated with the development of myocarditis [23]. In addition, the presence of actively replicating *PVB19* with detectable viral RNA [24] as well as the concomitant presence of lymphotropic viruses [25], may also be related to the occurrence of myocarditis, but this is less well established to date [26].

In Asia, a study by Chinese investigators showed that the prevalence of myocarditis was 11% in patients hospitalized with dengue infection. There were 201 out of 1782 total cases with myocarditis associated with dengue, and only 151 of them showed electrocardiographic (ECG) abnormalities, such as widened QRS complex, ST-T abnormalities, atrioventricular block and poor R wave progression [27].

A study in Australia including children younger than 10 years old found the annual incidence of cardiomyopathy to be approximately 1.24 per 100,000 children (95% confidence interval, 1.11 to 1.38). Lymphocytic myocarditis was an important cause of cardiomyopathy within this cohort (frequency 40%) [28].

Bacterial agents leading to myocarditis include pathogens such as *Corynebacterium diphtheria* [29], *Beta-haemolytic streptococci* [30], *Meningococci* [31], *Salmonella typhior paratyphi* [32,33], *Borrelia burgdorferi* [34,35,36], *Mycoplasma pneumonia* [37] and *Chlamydia psittaci* [38]. *Corynebacterium diphtheria* infection is now less frequent in western regions [39], while it is still a major public health problem in many underdeveloped countries and perhaps the most common cause of myocarditis worldwide [40]. *Βeta-haemolytic streptococcus* is the pathogen responsible for rheumatic fever. While rheumatic fever has a low prevalence in the western world and its incidence in the United States is less than 2 per 100,000 in the population, it is the leading cause of cardiac hospitalization in children and young adults in the age group of 5–25 years in developing countries [41]. *Chlamydia* is estimated to present with minimally symptomatic myocarditis in 5% to 15% of cases [38].

### 2.2. Toxicity or Hypersensitivity Reaction

Toxic myocarditis may be triggered by numerous agents. Prescribed drugs such as dobutamine, phenytoin [42], antibiotics (e.g., ampicillin, azithromycin, cephalosporins and tetracyclines), psychiatric medications (tricyclic antidepressants, benzodiazepines and clozapine) [43], recreational/illicit drugs (e.g., methamphetamine or cocaine) [44], heavy metals (copper, lead and arsenicals) and antineoplastic agents (e.g., anthracyclines, cyclophosphamide, 5-fluorouracil and tyrosine kinase inhibitors) are known etiologies of myocarditis [43]. With the increasing use of Immune-Checkpoint Inhibitors (ICIs) to treat a variety of cancers, reports of lethal myocarditis as potential adverse effects of therapy have increased. Mahmood and colleagues showed that the prevalence of myocarditis in patients treated with ICIs from 2014 to 2017 was 1.14%, with myocarditis developing approximately 34 days after the start of treatment with ICIs [45]. It was also found that incidence of myocarditis was higher in patients treated with ICI as compared to patients treated with other drugs that were included in the Vigibase database, a unique WHO global database including more than 16 million individual case safety reports submitted by national pharmacovigilance centers since 1967 [46].

Myocarditis was also reported as a hypersensitivity reaction during smallpox vaccination in the 1950s and 1960s [47,48]. The frequency of myocarditis was dependent on the strain used to produce the vaccine, and the method for detecting myocarditis. However, its true incidence was unknown. In the United States Military Dryvax vaccination program, which was a large-scale smallpox vaccination program, the incidence of myocarditis was estimated to be 0.01% or about one in 10,000 [48,49].

### 2.3. Autoimmunity

Autoimmunity has been increasingly recognized as one of the main factors sustaining inflammation and disease progression in myocarditis. Pathogens, such as viruses, may initiate autoimmune mechanisms that lead to inflammatory cardiomyopathy and myocarditis [50,51,52]. We recently showed that by the time of symptom onset of myocarditis, no relevant viral infection was detectable in patients with myocarditis when interrogated by full virome sequencing for all known vertebrate viruses. Importantly, this study also included myocardial tissue samples from patients with giant cell myocarditis (Figure 2). No relevant viral infection could be detected in those samples [53].

Additionally, systemic autoimmune diseases may involve the heart and manifest as myocarditis [54,55]. In that regard, it is estimated that cardiac involvement is present in 2% to 5% of patients with systemic sarcoidosis [56,57,58,59]. A study in Finland showed that between 1988 and 2012, the number of cases of cardiac sarcoidosis increased significantly, with the prevalence of cardiac sarcoidosis being 2.2 per 100,000 people in 2012 [60].

In patients with systemic lupus erythematosus (SLE), myocarditis may be a life-threatening manifestation. Although in clinical studies SLE myocarditis was found in only about 9% of patients with SLE [61,62,63,64], post-mortem analyses reported a high prevalence of 57%, indicating a high prevalence of subclinical disease [65,66,67]. SLE is most common in women in their 20s and 30s, and the severity as well as the clinical manifestations of the disease differ between early and late onset [68,69,70]. Specifically, it has been reported that patients diagnosed with SLE at an older age are more likely to develop a cardiovascular disease than patients diagnosed with SLE at a younger age [71]. However, it is noticeable that lupus myocarditis occurs more frequently in young people. While the prevalence in patients diagnosed with SLE before the age of 18 is 1%, the prevalence in elderly individuals is 0.3% [72].

Giant cell myocarditis is among the most aggressive forms of myocarditis and may be associated with autoimmune diseases, such as SLE, Sjögren’s syndrome, vasculitis, ulcerative colitis and polymyositis. Patients with giant cell myocarditis continue to have poor prognosis despite maximal therapy [73,74,75,76]. The epidemiology of giant cell myocarditis has not been investigated comprehensively given its low incidence. An autopsy study, including 377,841 autopsy cases, found the incidence of giant cell myocarditis to be 0.007% [77,78]. In comparison, the incidence of other types of myocarditis was considerably higher (0.11%) [78]. Another subtype of myocarditis in the context of systemic autoimmune disease is eosinophilic myocarditis, which is estimated to be present in 50–60% of cases of peripheral eosinophilia [79,80]. It has been shown that male sex hormones may act in a proinflammatory way, while female sex hormones protect against myocyte infectivity and reduce potentially harmful myocardial inflammatory response. Accordingly, eosinophilic myocarditis also occurs most commonly in males [81].

## 3. Epidemiology of Myocarditis

### 3.1. Prevalence of Myocarditis

Patients suffering from myocarditis are mostly male (82%), and young adults (average age: men: 40 ± 16; women: 40 ± 17) [82,83].

A study including 195 countries estimated 1.80 million (95% uncertainty interval (UI) 1.64 to 1.98) cases of myocarditis worldwide in 2017. The global number of deaths caused by myocarditis in 2017 was estimated to be around 46,486 (95% UI 39,709 to 51,824), and the highest age-standardized death rate was found in Oceania (2.6 (95% UI 2.0 to 3.4) per 100,000 people), most likely due to insufficient health resources in this region. The high-income Asia-Pacific region demonstrates the highest age-standardized prevalence of myocarditis (45.6 (95% UI 41.1 to 50.1) per 100,000 people). The prevalence rates of myocarditis varied between 10.2 (95% UI 9.0 to 11.4) per 100,000 people in Chile to 105.6 (95% UI 90.8 to 120.8) per 100,000 people in Albania (Figure 1). Across all 194 countries, prevalence rates for myocarditis differed by a factor of 10.4 and mortality rates by a factor of 43.9 in 2017. Between 1990 and 2017, age-standardized rates for myocarditis declined, while global prevalence and death rates increased significantly [84].

In several studies, myocarditis was found in 1.4–63% of EMBs from patients with unexplained congestive heart failure, unexplained ventricular arrhythmias or “primary” atrial fibrillation [85,86,87,88].Inflammatory changes were found in 22 of 35 patients with idiopathic congestive cardiomyopathy, suggesting that persistent myocardial inflammation may be associated with dilated cardiomyopathy (DCM) [88]. In line with those findings, myocardial inflammation is present in 13% to 27% of patients suffering from idiopathic DCM [89]. Moreover, observations from Western Europe and North America indicate that between 10% and 50% of acute DCM cases are likely to be associated with myocarditis [90].

### 3.2. Sex-Specific Differences

There is evidence for sex-specific differences in myocarditis regarding clinical, laboratory and pathophysiological features. Myocarditis is diagnosed more commonly in men than in women. Moreover, men are more likely to experience a severe trajectory of myocarditis, while women have a significantly lower risk of death or heart transplantation [91]. From 1990 to 2017, a greater decrease in age-standardized prevalence and mortality rates for myocarditis was observed in women vs. men [84]. It is important to note that clinical presentation appears to be more subtle in women than in men, leading to a potential underdiagnosis of myocarditis in women [82]. Additionally, male subjects showed a higher incidence of CVB3-induced myocarditis with a more severe clinical course than females in murine models [92]. Differences in the innate immune response to CVB3 between men and women could explain this phenomenon [93,94]. The latest advances in cellular and molecular biology have shown that both direct viral and immune-mediated injury are involved in the pathogenesis of enteroviral myocarditis [95]. While males present with increased γδ T cells; increased TLR4+ CD11b+ inflammation, including macrophages, neutrophils, mast cells and DCs; and an increased Th1 response, females present with protective Th2 response, increased B cells, more inhibitory Tim-3+ CD4+ T cells and more T regulatory cells [96,97].

### 3.3. Regional Differences in Myocarditis

#### 3.3.1. Infectious Myocarditis

Frequent infectious causes for myocarditis in Asia between 1966 and 2000 were associated with Diphtheria (Afghanistan and India) [98], typhoid fever [99] and viral infections, such as *CVB3* [100] and *Chikungunya*. *Hepatitis C virus* was suggested to be a significant cause of myocarditis in Japan, as its genomic material was detected in several cardiomyopathies [101,102,103]. Studies from Australia and New Zealand revealed regional differences with *CVB3* and *enterovirus 71* epidemics [104,105]. In Mexico, Central and South America, measles, meningococcal meningitis, *human immunodeficiency virus (HIV)*, dengue fever and diphtheria were characteristic etiologies of myocarditis, whereby Chagas disease (CD) was excluded from the survey [90]. Reports from Africa addressed *HIV* [106], peripartal cardiomyopathy and occasionally infections, such as trypanosomiasis [107] and shigellosis as some of the most common causes of myocarditis.

CD, induced by *Trypanosoma cruzi*, is a major cause of myocarditis in Latin America. It is estimated that 6–8 million people [108] are infected with *Trypanosoma cruzi* worldwide, with almost 11,000 deaths annually. EMBs showed myocardial inflammation in about 60% of CD patients [109]. Less than 1% of patients with acute CD have a severe course with acute myocarditis, pericardial effusion and/or meningoencephalitis, while the acute phase of CD is asymptomatic or includes non-specific clinical features [110]. In addition, while some newborns with congenital CD are asymptomatic, others may present with myocarditis, meningoencephalitis or hepatosplenomegaly. In particular, patients with AIDS who have been exposed to *Trypanosoma cruzi* are at risk for myocarditis, since immunosuppression may trigger reactivation of CD [111]. Due to migration from Latin America to Europe, the United States, Canada and Japan, CD has to be considered in patients worldwide. Most migrants from CD endemic areas are reported in Spain, Italy, France, the United States Kingdom and Switzerland [112,113,114,115]. Basile and colleagues estimated the expected number of infected people in Europe to be between 68,318 and 123,078 in 2009, whereas only 4290 cases of CD were diagnosed that year, suggesting that 94–96% were potentially undiagnosed cases [113].

#### 3.3.2. Toxic Myocarditis

Studies in Australia and New Zealand reported that the use of clozapine and anabolic steroids and, rarely, vaccination against smallpox are serious potential causes of toxic myocarditis [116,117,118]. Excessive alcohol use has been reported to be a common cause of toxic myocarditis. Thirty percent of EMBs of patients with excessive alcohol consumption showed lymphocytic infiltrates with myocyte degeneration and focal myocardial necrosis [119,120]. Based on data from 77 countries, alcoholic cardiomyopathy has been demonstrated in 1.54 billion adults, mainly in American and WHO European Regions [121].

Another cause of toxic myocarditis is iron overload with concomitant iron deposition in the myocardium and related myocardial injury. As a result, ventricular enlargement and dysfunction might occur, potentially leading to heart failure [122,123,124]. Patients with β thalassemia develop myocarditis, which is potentially involved in the pathogenesis of left ventricular systolic dysfunction. However, it has not been established, whether a direct association exists between myocarditis and iron overload [123].

Mesobuthus tamulus (Indian Red Scorpion) and Heterometrus swammerdami are species of a scorpion in India that may cause toxic myocarditis [125]. The scorpion bite causes a decrease in Na-K-ATPase and an increased myocardial oxygen demand by releasing adrenaline and noradrenaline from neurons, ganglia and adrenal glands, resulting in adrenergic myocarditis [126].

## 4. Role of CMR in the Diagnosis of Myocarditis

CMR is a sensitive and non-invasive tool for the diagnosis of myocarditis [127]. The “Lake Louise Criteria” for diagnosis of myocarditis were published in 2009 and updated in 2018. The initial Lake Louise Criteria consisted of T2 weighted imaging for the detection of edema, early and late gadolinium enhancement and achieved 67% sensitivity and 91% specificity in the diagnosis of myocarditis [8].Newer mapping techniques add substantial information regarding the presence of edema and improve diagnostic capabilities in regard to myocardial inflammation [128]. Moreover, CMR provides additional prognostic information and helps objectifying recovery and potential fatal consequences [129].

The implementation of EMBs, CMR and coronary angiography for the exclusion of CAD facilitated diagnosis of myocarditis. However, presentation of myocarditis often mimics CAD with angina-like symptoms and myocardial damage diagnosed by elevated troponin. As a result, patients with myocardial infarction and non-obstructive coronary arteries (MINOCA) represent an especially challenging population, as etiology and, therefore, therapeutic options are uncertain. As a result, MINOCA patients suffer from increased mortality and lack sufficiently targeted therapies [130,131].

Recent studies emphasize the relevance of CMR for the diagnosis of MINOCA [132,133]. Most importantly, myocarditis seems to be a common cause in this population. The prevalence of myocarditis in patients with MINOCA was found to be between 15% and 75% [134,135,136,137,138,139].

A study by Pasupathy and colleagues found that 33% of patients with MINOCA had myocarditis, utilizing CMR for diagnosis [130]. We demonstrated that a systematic use of CMR in patients suffering from MINOCA increased the overall incidence of myocarditis up to 6.2-fold in a tertiary care center in Switzerland [132,133]. This demonstrates the importance of developing systemic diagnostic algorithms and unveils an approximation of the true epidemiologic significance of myocarditis. Certainly, as mentioned above, a definitive diagnosis can only be made with EMB.

From 2020 onwards, the current NSTEMI guidelines recommend CMR in all patients with a first diagnosis of MINOCA. This may lead to higher numbers of reported cases of myocarditis in the future.

## 5. Updates on SARS-CoV-2 Infection and COVID-19 Association with Myocarditis

SARS-CoV-2 infection and its resultant clinical manifestation COVID-19 rapidly spread all over the world in 2020. Interestingly, the first studies reported cardiac injury, defined by increased levels of troponin, in 22.2–31% of COVID-19 patients. Troponin elevation was associated with severe COVID-19 infection, increased morbidity and mortality [140,141,142]. These results were later confirmed in a larger case series of 5700 patients [143]. Moreover, increased serum troponin levels were associated with higher in-hospital mortality [144,145]. However, most early studies neither included specified characteristics of patients with increased troponin nor further cardiac imaging or EMB. As a result, the cause of troponin elevation remains uncertain. As myocarditis was already discussed to be a potential complication of middle east respiratory syndrome (MERS) and other *coronaviruses*, a link between COVID-19 and myocarditis seems rational [7,146]. Nevertheless, data on this topic are still quite preliminary.

There are numerous case reports associating COVID-19 and myocarditis [147,148,149,150]. A recent study by Puntmann and colleagues detected cardiac involvement in 78 patients (78%) and persistent myocarditis in 60 patients (60%) by CMR after COVID-19 infection [151]. Patients that had recently recovered from COVID-19 presented lower left ventricular ejection fraction, higher left ventricular volumes and raised native T1 and T2 than healthy controls and risk-factor matched controls. In total, 78 patients (78%) revealed abnormalities in CMR, such as raised myocardial native T1 (*n* = 73), raised myocardial native T2 (*n* = 60), myocardial late gadolinium enhancement (*n* = 32) or pericardial enhancement (*n* = 22). Pre-existing conditions, severity and overall course of the acute disease and time since original diagnosis had no impact on the results [151]. These data have to be interpreted with caution, since definitive diagnosis of myocarditis via EMB was only obtained in a subset of patients.

A CMR study of 26 athletes with COVID-19 with mild or no symptoms revealed findings suggestive for myocarditis in 4 of them (15%) and late gadolinium accumulation without T2 elevation indicating previous myocardial injury in 8 other athletes (30.8%) [152].

A recent autopsy study by Halushka et al. reported a much lower incidence of myocarditis in patients who died from COVID-19. Out of 277 autopsies, only 20 showed signs of myocarditis, which represents only 0.07% of patients who died from COVID-19 in this cohort [153].

Other cardiovascular complications appear to be more predominant in patients with COVID-19. Thrombotic events, arrhythmias as well as aggravation of an underlying heart disease may also result in myocardial damage, potentially worsening prognosis [154]. An autopsy study reported that only 5 out of 68 (7%) fatal cases of COVID-19 were associated with myocardial damage and circulatory failure [155].

Moreover, it has been suggested that underlying cardiovascular disease may promote a more severe course of COVID-19 infection [141,142,155,156,157,158]. In this context, a meta-analysis of 1527 COVID-19 patients showed that 17.1% of these patients were hypertensive and 16.4% had pre-existing heart disease [159]. In another study involving 44,672 COVID-19 patients, the fatality rate in patients with cardiovascular disease was found to be five times higher than in patients without underlying cardiovascular disease [157].

Another recent study including 47 heart transplant recipients showed that there was a high rate of death after infection with SARS-CoV-2 in this patient group compared with the general population. The higher death rates were attributable to comorbidities. No evidence of myocardial injury was found in any of these patients, suggesting that cardiac involvement by COVID-19 may be rare among cardiac transplant recipients [160].

In summary, the association of COVID-19 and acute cardiac injury has been repeatedly reported and is associated with poor outcomes. Data on myocarditis in patients with COVID-19 have been somewhat controversial, although the overall consensus is that myocarditis is still rare in this patient population. Whether myocarditis will reveal itself to be causative or a bystander in the development of complications of COVID-19 remains an area of research. Indeed, with regard to the proportions of the COVID-19 pandemic, we speculate that there may be a wave of cardiac post-COVID-19 complications in the long run [161].

## 6. Conclusions

The prevalence of myocarditis has been estimated to range from 10.2 to 105.6 per 100,000 people worldwide, with relevant regional differences influenced by a variety of pathogens, as well as locally available diagnostics procedures. Based on studies performed in 2017, the annual prevalence of myocarditis is about 1.8 million worldwide. While we slowly approximate the true incidence of myocarditis, its true proportions remain unknown.

In this context, the establishment of CMR as a standard for the diagnosis of myocarditis provides the possibility of rapid detection of myocarditis before complications might occur. Although myocarditis is associated with high morbidity and mortality [162,163,164,165], clinical practice recommendations remain vague due to the lack of data from large clinical trials [166]. The limited awareness of this disease leads to an even deeper knowledge gap that we urgently need to fill. Considering the appearance of novel pathogens, such as SARS-CoV2, an increasing expansion of pathogens, such as CD, through globalization in combination with improved diagnostic tools, we expect a further increase in the incidence of myocarditis over the next decade. International collaborations will be necessary to make progress in developing effective therapies for this increasing patient population.

## Figures and Tables

**Figure 1 jcm-10-00603-f001:**
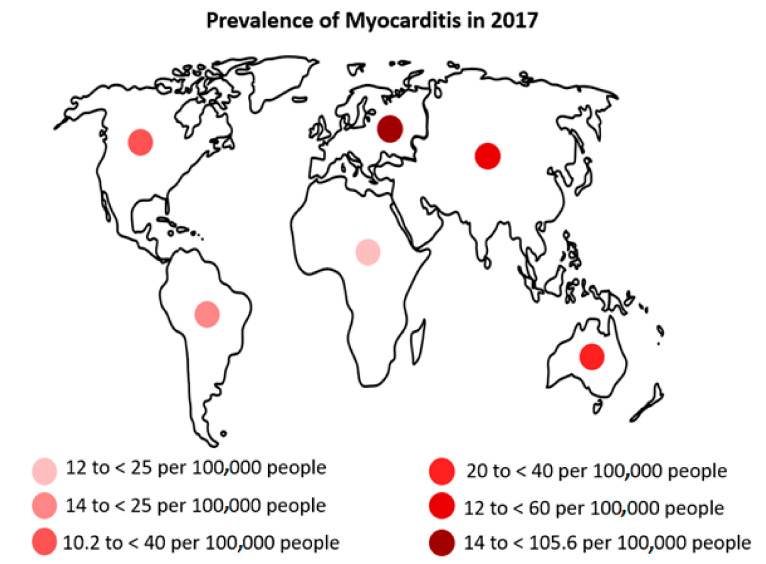
Age-standardized prevalence of myocarditis in 2017.

**Figure 2 jcm-10-00603-f002:**
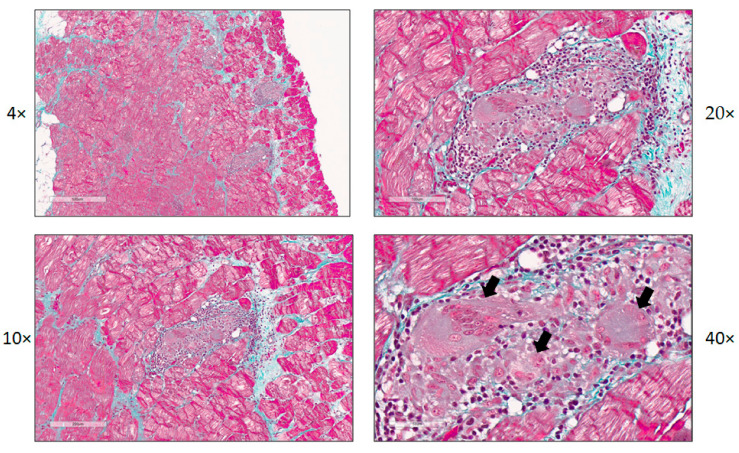
Histology and immunohistochemistry of a patient with giant cell myocarditis. Images show a lesion of giant cell myocarditis (magnification 4×–40×). Masson’s trichrome staining (panels 1–4), arrows point out a giant cell and cells in the process of forming a giant cell.

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
