# Peer review of "Epidemiological Impact of Myocarditis"

_jcm, 2021, doi:10.3390/jcm10040603_

Round 1

Reviewer 1 Report

The paper by Golpour and colleagues provides an extensive review on myocarditis. The paper is well written and well-structured describing the various etiologies and the epidemiology of myocarditis as well as the utility of CMR as a diagnostic tool. In addition, it includes a chapter on COVID-19-associated disease which further increases the timeliness of the paper.

As two small points, authors could mention or discuss in a few sentences the value/benefit of electroanatomical-mapping guided endomyocardial biopsy in establishing the diagnosis of myocarditis. On page 4, the sentence starting with “systematic review and meta-analysis showed that 22 (25%)…..” would require a citation. In addition, it is unclear what the 25% refers to and would need to be clarified.

Author Response

Dear associate editor,

Thank you very much for your feedback.  We are very happy that we were able to adequately address the literature on the epidemiology of myocarditis.  Please find attached our point-by-point response to the statistical review.  In addition, we have made minor changes with track tool within the manuscript to improve writing style and clarity.  The content of the manuscript has not been changed.

Reviewer 2 Report

Epidemiological impact of Myocarditis

The authors do an excellent job of reviewing the literature on myocarditis. This kind of synthesis work is very useful. It is very well written and structured.

I only recommend adding in session 5 (SARS-CoV-2 infection and myocarditis), a recent work published in JACC Heart Failure. This work summarizes the world's largest experience of heart transplant patients who have been infected with SARS-CoV-2. Myocarditis was not recognized in any of these patients, but the fatality rate was significantly higher than in the general population, due to the comorbidities.

This work confirms that SARS-CoV-2 cardiac involvement is rare.

COVID-19 in Heart Transplant Recipients: A Multicenter Analysis of the Northern Italian Outbreak. Bottio T, et al. JACC Heart Fail. 2021 Jan;9(1):52-61. doi: 10.1016/j.jchf.2020.10.009. Epub 2020 Oct 29.

PMID: 33309578

Author Response

(The authors gave the same response as above.)

Reviewer 3 Report

Dear Authors

Study is interesting but COVID-19 and myocarditis relation you can only write if you have some studies of incidence of myocarditis in 2020 onwards.

So please add those studies if you wish to compare Pre-2020 vs 2020 Prevalence.

Thanks

Author Response

(The authors gave the same response as above.)
